# OpenReview forum: "AdAEM: An Adaptively and Automated Extensible Measurement of LLMs' Value Difference"
_ICLR.cc/2026/Conference — ICLR 2026 Oral_

### Official Review · Reviewer_xTVE · 2025-10-25

**Soundness:** 4
**Presentation:** 3
**Contribution:** 3
**Rating:** 8
**Confidence:** 3

**Summary:**

This paper proposes AdAEM, a dynamic and self-extensible framework for evaluating LLM value orientations, addressing the "informativeness challenge" in existing static benchmarks. By adaptively generating test questions that maximize value distinctions across different LLMs, AdAEM constructs an evaluation benchmark containing 12,310 questions and systematically evaluates 16 mainstream LLMs.

**Strengths:**

- The paper clearly identifies a critical issue in existing value evaluation methods—using outdated, contaminated, or overly generic test questions leads to indistinguishable and uninformative results. This is a genuine and important research problem.

- The optimization algorithm is grounded in an information-theoretic objective (Equation 1) and employs an EM-like algorithm to alternately optimize question generation and response selection.

-  The method is evaluated on 16 mainstream LLMs covering different scales, architectures, and regions, validated across multiple dimensions (question quality, validity, reliability, extensibility).

**Weaknesses:**

- Reliability of probability estimation: For black-box LLMs, off-the-shelf classifiers approximate p(v|y) and coherence measures approximate p(y) (Section 3.2), but error propagation from these approximators to the final optimization objective is insufficiently discussed, potentially introducing systematic bias.

- Limited Evaluation Scope: Only instantiated on Schwartz's value system; while Appendix H acknowledges limitations, no validation experiments on other value theories (e.g., MFT, Hofstede) are provided, limiting generalizability.

- Reproducibility Considerations: Selection criteria for P1 and P2 models unclear: Why these 9 specific models? Would replacing with other model combinations significantly affect generated question quality?

**Questions:**

-  When comparing with ValueDCG (4,561 questions) and ValueBench (40 questions), does AdAEM's 12,310 questions introduce a scale advantage? Can you provide fair comparison with controlled question counts?

 - What are the background distributions of the 5 experts (nationality, expertise)? The 300 samples represent only 2.4% of the total dataset—how do you ensure evaluation result representativeness?

---

> ### Author Response · Authors · 2025-11-22
> **Thanks very much for your valuable suggestions. Response to Reviewer xTVE (1/3)**
>
> Thank you very much for your supportive review and valuable suggestions. Our responses to each Weaknesses (W) and Question (Q) are as follows. A revision of our paper has been uploaded, with the revised parts marked in blue.
>
> ---
> ### **W1: Reliability of probability estimation: off-the-shelf classifiers approximate p(v|y) and p(y) may introduce systematic bias.**
>
> To ensure AdAEM is compatible with black-box LLMs where exact generation probabilities are inaccessible, we introduce the empirical approximation to estimate the probability and divergence, as detailed in Appendix C.3.
>
> **The good quality, reliability, and validity of AdAEM-generated questions** (verified in Sec.4.2, Appendix C.9, C.10, C.11, J, K) **have demonstrated such approximation is empirically acceptable**.
>
> To further address this concern, we add another experiment and implement Eq.(1) strictly following the derived mathematics using open-source LLMs (P1 = P2 = { LLaMa-3.1-8B, Qwen2.5-7B, Mistral-7B-v0.3}), and compare the questions generated via approximated (defined as $X_{\text{appro}}$) and exact (defined as $X_{\text{exac}}$ ) Eq.(1), respectively.
>
> We compared LLM value orientations obtained with $X_{\text{appro}}$ and $X_{\text{exac}}$, respectively, and observed:
> + Pearson correlation=0.86, suggesting that **the approximated Eq.(1) produces similar results as the exact Eq.(1)**.
> + Cronbach's alpha=0.90, indicating that **the approximated Eq.(1) measures the same constructs as the exact one**.
>
> This empirical comparison shows that **our approximations preserve the effectiveness of the method** and would not introduce systematic bias.
>
> Please refer to Appendix I.2 for more details and full experimental results.
>
> ---
>
> ### **W2: No validation experiments on other value theories are provided, limiting generalizability.**
>
> To further validate generalization capability, we also instantiated AdAEM with the **Moral Foundation Theory (MFT)**. We summarize the key conclusions on MFT below. Please refer to **our response to Reviewer SGLU's W1 and Appendix J** for more detailed results. Thank you!
>
> + **Question Quality**: **AdAEM generates questions for MFT with much higher semantic diversity** than manually crafted MFT questionnaires or ValueBench.
> + **Value Difference Elicitation Ability**: We also check AdAEM's capability of exposing LLMs' underlying value difference on MFT, by evaluating *GPT-4.1, Mistral-7B-v0.3, Llama-3.3-70B-Instruct, and DeepSeek-V3* with AdAEM Bench-MFT, Moral Foundations Questionnaire (MFQ) and ValueBench. Define $v_{\text{LLM}}^{\text{Method}}$ as the obtained value orientations by each method, e.g., $v^{\text{MFQ}}_\text{GPT} \in \mathbb{R}^5$.
>     + Evaluated by MFQ or ValueBench, the average Pearson correlation of values among different LLMs, e.g., corr($v_\text{GPT}^{\text{MFQ}}, v_\text{DS}^{\text{MFQ}}$), is ~0.6, indicating *different models' value tendencies are implausibly similar* measured by these two methods.
>     + Evaluated by MFQ or ValueBench, the average standard deviation of LLM's tendency scores across the five foundations, e.g., std($v_\text{Mistral}^{\text{VB}}$) is quite low (~0.1), indicating that *neither of them successfully reveals LLM value differences*.
>
>     In comparison, **AdAEM leads to low correlation of values among different LLMs (Pearson=$-0.1$) and high distinguishability across values (std=0.21)**, better exposing more value differences and providing informative results. (Refer to Table 16 and Fig.19 in Appendix J.2 for detailed results and radar chart visualization).
>
> + **Evaluation Validity**: To verify that AdAEM Bench can reflect the real values of LLMs, we also conduct the value priming experiment using GPT-5, as we did in Sec.4.2. Results show **AdAEM successfully captures the significant increase towards the target foundation**.
>
> **The results above show that AdAEM generalizes effectively to Moral Foundation Theory, preserving its key advantages** to generate diverse, value-evoking questions to uncover LLMs' value differences, further supporting that AdAEM is a general framework rather than one tied to a particular value theory.
>
> ---
>
> ### **W3: Selection criteria for P1 and P2 models are unclear. Would replacing with other model combinations significantly affect generated question quality?**
>
> **Criteria for selecting P1 and P2**. AdAEM aims to optimize value-eliciting questions by exploiting up-to-date and cultural LLMs' knowledge and value boundaries, to address the informativeness challenge. The **key requirement lies in the potential diversity of their underlying values**. Driven by this, as stated in Sec. 4.1, we choose LLMs that are developed in different cultures and time periods. To enhance reproducibility, we have made this selection criterion explicit in Sec. 4.1 and Appendix. C.1, of the revision.

---

> ### Author Response · Authors · 2025-11-22
> **Thanks very much for your valuable suggestions. Response to Reviewer xTVE (2/3)**
>
> **AdAEM's robustness to P1 and P2**. To further demonstrate robustness in model combinations, we implement AdAEM with an alternative set of LLMs (P1={ LLaMa-3.1-8B, Qwen2.5-7B, Mistral-7B-v0.3,GPT-4o-Mini}=P2)  and compare the generated questions (denoted as AdAEM-2) with the original AdAEM Bench. We summarize the key results below. Please refer to **our response to Reviewer J1nE's W4 and Appendix K.1 for details**.
>
> + *Question quality*: AdAEM and AdAEM-2 produce questions with greater semantic diversity and topic richness than the manually crafted SVS.
> + *Informativeness score*: As shown below, AdAEM and AdAEM-2 significantly improve the score/reward over the initial general questions.
>
>     | **Dataset**     | **General** | **AdAEM** | **AdAEM-2** |
>     |-----------------|-------------|-----------|-------------|
>     | **Mean Reward** | 6.07        | **6.99**      | *6.51*  |
>
> + *Correlation of evaluation results*: Using both sets to evaluate the values of LLMs, we compute the agreement between their evaluation results using four metrics: Intra-class Correlation (ICC), Pearson Correlation, Spearman Correlation, and Cronbach's $\alpha$ coefficient. As shown below, the high correlation demonstrates *AdAEM can produce consistent value assessments across model combinations*.
>     | **Metric** | **ICC** | **Pearson Corr** | **Spearman Corr** | **Crobach** |
>     |------------|---------|------------------|-------------------|--------|
>     | **Value**  | 0.82  | 0.79           | 0.73            | 0.84 |
>
> In summary, when the criteria of selecting P1 and P2 are satisfied, AdAEM's outcomes are robust to the choice of underlying LLMs.
>
> ----
>
> **Q1: When comparing with ValueDCG (4,561 questions) and ValueBench (40 questions), does AdAEM's 12,310 questions introduce a scale advantage? Can you provide fair comparison with controlled question counts?**
>
> 1.**The ability to generate extensive questions is AdAEM's unique strength**. AdAEM is designed to automatically expand from a small set of general topics and iteratively generate value-evoking questions. Unlike manually created or fixed benchmarks, generating a larger number of more informative questions that better uncover LLMs' value differences is an inherent and core advantage of AdAEM.
>
> 2.**Fair comparison with the same count**. To further address this concern, we randomly sample 200, 500, and 1,000 questions from the full 12,310-item set and compare the results with that of the full version. As shown below, even 200 samples yield consistent results with the full benchmark, and 1000 samples show strong consistency, which have comparable or smaller scale than ValueDCG (4,561) and ValueBench (40).
>
> | **#q**   | **ICC** | **Pearson** | **Spearman** | **Cronbach** |
> |----------|---------|-------------|--------------|--------------|
> | **200**  | 0.67   | 0.61       | 0.63        | 0.79        |
> | **500**  | 0.71   | 0.67       | 0.69        | 0.75        |
> | **1000** | **0.78**   | **0.69**       | **0.72**        | **0.84**        |
>
> Besides, we also compare the question quality with 1000 questions from AdAEMBench below, and still obtain good results.
>
> | **Dataset**   | **#q** | **Avg.L $\uparrow$** | **SB $\downarrow$** | **Sim $\downarrow$** |
> |---------------|--------|-----------|--------|---------|
> | **SVS**        | 57     | 13.00     | 52.68  | 0.61    |
> | **VB**         | 40     | 15.00     | 26.27  | 0.60    |
> | **DCG**        | 4,561     | 11.21     | 13.93  | 0.36    |
> | **AdAEM-full** | 12,310 | 15.11     | 13.42  | 0.44    |
> | **AdAEM-1,000** | 1,000  | 16.17     | 13.95  | 0.47    |
>
> These results demonstrate that AdAEM not only offers the advantage of automatically generating scalable evaluation items, but also its generated items are of sufficiently high quality to ensure robust and reliable evaluation under limited question count.
>
> We added this analysis in Appendix K.2.
>
> ---

---

> ### Author Response · Authors · 2025-11-22
> **Thanks very much for your valuable suggestions. Response to Reviewer xTVE (3/3)**
>
> **Q2: What are the background distributions of the 5 experts (nationality, expertise)? The 300 samples represent only 2.4% of the total dataset—how do you ensure evaluation result representativeness?**
>
> We have provided the background details of the 5 human experts in Appendix C.10: "Two English-proficient graduate students with social science backgrounds and three external social science experts (recruited via an open call). None of the authors advise, supervise, teach, or evaluate these students; no hierarchical relationship exists."
>
> Regarding the representativeness of the 300 evaluated samples (2.4% of the dataset), we acknowledge the importance of large-scale human evaluation, but we'd like to offer the following clarifications:
>
> 1. **The current number of annotators and samples aligns with common practice in LLM/NLP research**. Many existing studies on values rely on similar or even smaller-scale human evaluations [1][2]. While we agree that a larger scale would further strengthen our findings, it is often impractical due to the high cost of human labor.
> 2. **AdAEM is robust to question counts**. As shown in our response to Q1, even random subsets of 200 or 500 questions can yield certainly consistent results with the full 12,310-item benchmark. This also confirms the representativeness of a randomly sampled subset of 300 samples to the total dataset.
> 4. **The human evaluation only serves as one dimension of our benchmark validation**. We have verified the dataset's validity through multiple complementary approaches, including question quality comparison, validity analysis, and reliability analysis in Sec.4.2, Appendix C.9, C.10, C.11, J, K.
>
> ---
>
> **Flag For Ethics Review: Privacy, security and safety; Potentially harmful insights, methodologies and applications; Responsible research practice (e.g., human subjects, annotator compensation, data release)**
>
> We appreciate the reviewer's attention to ethical considerations. We would like to clarify how our submission explicitly addresses each raised concern:
> + **Safety, potentially harmful insights, methodologies, and applications**: As detailed in our Ethics Statement (page 10), we explicitly acknowledge that a framework capable of generating controversial questions could be potentially misused. We emphasize:
> (1)	Our framework is not designed for any malicious purposes but to improve the alignment and ethical development of LLMs. This needs to engage LLMs with sensitive content to uncover potential biases and value-associated risks, which is a necessary step towards reducing real-world harms.
> (2)	To mitigate social harms, we have already detected and removed any evaluation questions that could elicit seriously harmful outputs, preventing direct social harm.
> + **Human subjects & annotator compensation**: Appendix C.10 provides full details on human evaluation, including the annotation process, the background information of annotators, compensation, and time accounting. Importantly, all annotators were paid 12 USD per hour, 41% above the local minimum wage of 8.50 USD per hour.
>
> We believe these safeguards and transparent reporting sufficiently address the ethical concerns and reflect a strong commitment to responsible research.
>
> ---
>
> ### Reference
> [1] Ren et al., ValueBench: Towards Comprehensively Evaluating Value Orientations and Understanding of Large Language Models. ACL 2024.
>
> [2]  Sorensen et al., Value Kaleidoscope: Engaging AI with Pluralistic Human Values, Rights, and Duties. AAAI 2025.

---

> > ### Comment · Reviewer_xTVE · 2025-11-27
> >
> > Thanks for the explanation, which clears some of my concern. I have no other concerns.

---

### Official Review · Reviewer_1aVD · 2025-10-30

**Soundness:** 4
**Presentation:** 4
**Contribution:** 4
**Rating:** 8
**Confidence:** 4

**Summary:**

This paper introduces AdAEM, a novel, dynamic, and self-extensible framework for evaluating the value differences in LLMs. The authors correctly identify the "informativeness challenge" in current static benchmarks, which often fail to distinguish between models because they test for generic, shared safety values. AdAEM tackles this by automatically generating new, controversial, and timely test questions by probing the value boundaries of diverse LLMs. This dynamic approach allows the benchmark to co-evolve with LLMs, mitigating data contamination and providing a much-needed tool for assessing misalignment, cultural bias, and preference.

**Strengths:**

The paper addresses a critical and well-defined problem in LLM evaluation. Moving from static to a dynamic, self-extensible benchmark (AdAEM) is a significant conceptual and practical contribution.

The core idea of using in-context optimization to probe value boundaries across different cultures (diverse LLMs) and time periods (knowledge cutoffs) is clever and highly effective at generating informative questions.

The authors provide a thorough empirical analysis. The results clearly show that AdAEM yields evaluation results that are significantly more distinguishable than existing benchmarks (ValueBench, ValueDCG).

The controlled value priming experiments provide strong support for the benchmark's construct validity, showing it can accurately detect intended value shifts.

**Weaknesses:**

The framework's current implementation relies solely on Schwartz's Theory of Basic Values. While well-justified, this is just one of many value frameworks, and the paper could benefit from a brief discussion on how AdAEM could be adapted to other theories (e.g., Moral Foundations Theory).

The paper acknowledges the potential for misuse, but it is worth noting that a framework designed to find controversial topics could indeed be misused.

**Questions:**

N. A.

---

> ### Author Response · Authors · 2025-11-22
> **Thank you very much for your supportive review. Response to Reviewer 1aVD**
>
> Thank you very much for your supportive review and valuable suggestions. Our responses to each Weakness (W) are as follows. A revision of our paper has been uploaded, with the revised parts marked in blue.
>
> ---
> ### **W1: The framework's current implementation relies on just one value framework, which benefits from discussion on how AdAEM could be adapted to other theories**.
>
> Thanks for the suggestion. To further validate generalization capability, we also instantiated AdAEM with the **Moral Foundation Theory (MFT)**. We summarize the key conclusions on MFT below, and please refer to **our response to Reviewer SGLU's W1 and Appendix J** for more details.
>
> + **Question Quality**: **AdAEM generates questions for MFT with much higher semantic diversity** than manually crafted MFT questionnaires or ValueBench.
> + **Value Difference Elicitation Ability**: We also check AdAEM's capability of exposing LLMs' underlying value difference on MFT, by evaluating *GPT-4.1, Mistral-7B-v0.3, Llama-3.3-70B-Instruct, and DeepSeek-V3* with AdAEM Bench-MFT, Moral Foundations Questionnaire (MFQ) and ValueBench. Define $v$ (Method, LLM) $\in \mathbb{R}^5$ as the value orientations of each LLM obtained by each method, e.g., $v$ (MFQ, GPT-4.1).
>     + Evaluated by MFQ or ValueBench, the average Pearson correlation of values among different LLMs, e.g., corr($v$ (MFQ,GPT-4.1), $v$ (MFQ, DeepSeek)), is about **~0.6**, indicating *different models' value tendencies are implausibly similar* measured by these two methods.
>     + Evaluated by MFQ or ValueBench, the average standard deviation of LLM's tendency scores across the five foundations, e.g., std($v$ (ValueBench, Mistral)) is **quite low (~0.1)**, indicating that *neither of them successfully reveals LLM value differences*.
>
>     In comparison, **AdAEM leads to low correlation of values among different LLMs (Pearson=$-0.1$) and high distinguishability across values (std=$0.21$)**, better exposing more value differences and providing informative results. (Refer to Table 16 and Fig.19 in Appendix J.2 for detailed evaluation results and radar chart visualization).
>
> + **Evaluation Validity**: To verify that AdAEM Bench-MFT can reflect the real values of LLMs, we also conduct the value priming experiment using GPT-5, as we did in Sec.4.2. Results in the following table show **AdAEM successfully captures the significant increase towards the target foundation**.
>
>     | **Dimension** | **Baseline** | **Controlled** | **Improvement** |
>     |---------------|--------------|----------------|-----------------|
>     | **Care**      | 74.88        | 98.31          | 31.29%          |
>     | **Fairness**  | 80.43        | 98.07          | 21.93%          |
>     | **Loyalty**   | 54.35        | 98.79          | 81.77%          |
>     | **Authority** | 57.25        | 98.07          | 71.30%          |
>     | **Sancity**   | 30.19        | 97.83          | 224.05%         |
>
> **The results above show that AdAEM generalizes effectively to Moral Foundation Theory, preserving its key advantages** to generate diverse, value-evoking questions to uncover LLMs' value differences.
>
> Please refer to our response to Reviewer SGLU's W1, as well as Appendix. J for more details. Thank you!
>
> ---
> ### **W2: A framework designed to find controversial topics could indeed be misused.**
> Thanks for your valuable suggestion. Discussion about the potential for misuse has been included in the *Ethics Statement (page 10) of the initial submission*. We emphasize two key points here:
> + Our intention of designing this framework is not for any malicious purposes but to improve the alignment and ethical development of LLMs, where it is necessary to engage LLMs with sensitive content to uncover potential biases and value-associated risks.
> + To mitigate social harms, we have already reviewed and removed the test questions that could elicit seriously harmful outputs.
>
> For the final version, we will enforce stronger safeguards to clear any questions that can produce potentially sensitive model outputs.
>
> We added more discussions in the Ethics Statement in the updated version.

---

### Official Review · Reviewer_J1nE · 2025-10-31

**Soundness:** 2
**Presentation:** 3
**Contribution:** 3
**Rating:** 4
**Confidence:** 3

**Summary:**

The paper proposes AdAEM, an information-theoretic framework that automatically and adaptively generates and expands test items to measure differences in LLM value orientations. The core method optimizes a generalized JS-divergence with disentanglement regularizer and produces “high controversy, model discriminative” value-inducing questions via sampling and refinement loops. Within this framework, the authors also build AdAEM Bench with 12,310 items that have been validated on quality, diversity, validity, and reliability.

**Strengths:**

Rigorous, scalable measurement of LLM value orientations is a timely, high-impact question for safety, alignment, and governance communities. This paper provides a dynamic, adaptive evaluation of LLM values, helping to overcome the limitations of traditional static benchmarks—namely their inability to capture new events, the difficulty of updating datasets dynamically, and the reliance on manual maintenance.
The central claims and proposed contributions are supported by well-organized benchmark statistics and visualizations, along with human validation on reasonableness and value differentiation. Construct validity is evidenced via controlled value priming with significant shifts in target/opposite dimensions, and reliability is supported by five-fold internal consistency. Contrast and contextualization against existing benchmarks and prior works are adequate.

**Weaknesses:**

1. Methodologically, selecting questions that maximize divergence in LLM responses does not necessarily best reveal their value preferences; it may instead induce new biases. This could be a significant shortcoming of the work and requires careful justification.
2. The EM/IM-like alternating procedure lacks a formal convergence or monotonic guarantee in the main text.
3. Too much empirical approximation of mathematical modeling injures solidity.
4. Any reasons on your choice of hyper-parameters? No experiments or searches provided.

**Questions:**

1. Figure 2 can be polished as it directly provides an overall illustration of AdAEM framework at the very beginning.
2. It would be better to unify the notation in main text, notation table and pseudo code, strictly avoiding the occurrence of unexplained and nonuniform symbols.
3. The “Figure 15” in AdAEM Question Generation on page 23 probably should be “Figure 12”.

---

> ### Author Response · Authors · 2025-11-22
> **Thank you for your insightful comments and constructive feedback. Response to Reviewer J1nE (1/3)**
>
> Thank you very much for your insightful comments and constructive feedback. We provide responses to each Weakness (W) and Question (Q) as follows. A revision of our paper has been uploaded, with the revised parts marked in blue.
>
> ---
> ### **W1: Question selection to maximize response divergence does not best reveal their value preferences and may instead induce new biases**.
>
> Note that our claimed motivation is **NOT** fully revealing an LLM's value preference but rather to exploit the parts where different LLMs' values **diverge**, as claimed in L16, 48-49, 82-83, 149, etc. A detailed discussion about our motivation was explained in Appendix G of the initial submission.
>
> To elaborate on this, we further explain it as below:
>
> **Core motivation of AdAEM**: LLMs' value preferences can be divided into two categories: i) **universal values** shared across communities, e.g., fairness, security, and privacy protection; ii) **unique values** distinctive from each other, e.g., cultural values and norms in different regions. Based on this, we have two observations:
>
> 1. **Previous benchmarks can already evaluate LLMs' orientations on universal values well enough**. As a typical RAI practice, most mainstream LLMs have been well aligned with these universal values, e.g., fairness, through the alignment process [1,2]. As a result, current benchmarks [3,4,5] can easily capture such salient preferences. However, this also often makes the revealed value scores of different LLMs saturated and indistinguishable. For example, in Fig.9 (a), ValueBench shows both GPT-4-turbo and Mistral-Large obtain almost full scores on *Security* and *Self-Direction*. We get *NOT any actionable insights from such results* to help users' model selection and diagnosis, i.e., *the informativeness challenge* (L46).
> 2. **Existing methods fail to reveal LLMs' value difference**. Due to differences in architecture, data, and training method, discrepancies are expected both across models and across value dimensions within the same model. However, existing benchmarks fail to capture these differences. For example, the correlations of values across different LLMs measured by MFQ and ValueBench appear implausibly high (Pearson's r~0.6, see details in Fig. 9(a) and our response to W1 of Reviewer SGLU).
>
> Therefore, we aim to complement current evaluations of well-aligned safety-related values and reflect more value differences (L48). In other words, **exposing the value bias of LLMs is exactly our motivation**.
>
> Besides, we'd like to emphasize:
>
> (a) **No additional bias is introduced**. Though we conduct question selection, **all models are evaluated on the same set of questions**, ensuring a fair and direct comparison. For the same question, distinct expressed preferences by LLMs actually reflect their inherent value differences, insetad of biased item construction.
>
> (b) **AdAEM has demonstrated good evaluation reliability and validity**. We have conducted extensive validity analysis in Sec.4.2, Appendix C.11, and Appendix. K, showing AdAEM Bench serves as a valid value measurement.
>
> Therefore, we believe our method uncovers real value differences of LLMs. The evaluation of shared values is out of our scope.
>
>
> Please kindly let us know if you have other questions/concerns.
>
> ---
> ### **W2: The EM/IM-like alternating procedure lacks a formal convergence or monotonic guarantee in the main text.**
> We clarify this issue from two perspectives:
>
> + **Theoretical support**. Our procedure directly follows the classical Information Maximization (IM) framework, which alternately optimizes a variational lower bound of the objective in Eq. (1). This EM-like alternating optimization is a well-established approach for iteratively tightening the lower bound and moving toward the objective. Its *convergence has been proved in Proposition 2.1 of [6]*, where it is shown this family of methods "is guaranteed to maximize or leave unchanged a lower bound on the mutual information".
>
> + **Empirical evidence**. In our original Optimization Efficiency analysis part (Fig.7, Sec.4.3), we have empirically demonstrated that AdaEM's optimization can monotonically increase (with slight fluctuations) the scores of the generated questions. To further validate this property, **we add a small-scale experiment in Appendix. E.6** (Fig.18) with $B=100$, and **observed the scores monotonically increase and stabilize** after more iterations, **converging to a maximum** higher than the initial one.
>
> We added this convergence discussion in Appendix. E.6.
>
> ---
> ### **W3: Too much empirical approximation of mathematical modeling injures solidity.**
>
> Our method introduces two kinds of approximation for computational tractability and practical feasibility. We clarify why these approximations are necessary, acceptable, and do not influence AdAEM's effectiveness below:

---

> ### Author Response · Authors · 2025-11-22
> **Thank you for your insightful comments and constructive feedback. Response to Reviewer J1nE (2/3)**
>
> + **Mathematical approximation to obtain a tractable bound**, including i) lower bound of divergence and ii) Monte Carlo approximation for expectations and sampling. *All these approximations are widely used common practice in the divergence and mutual information estimation or maximization* [7,8]. Please refer to Appendix I.1 for the detailed explanation.
>
> + **Empirical approximation** to ensure AdAEM is compatible with black-box LLMs where exact generation probabilities are inaccessible, as detailed in Appendix C.3. The good quality, reliability, and validity of AdAEM-generated questions (verified in Sec.4.2, Appendix C.9, C.10, C.11, J, K) **have demonstrated such approximation is empirically acceptable**.
>
> To further address this concern, **we implement Eq.(1) strictly following the derived mathematical form with open-source LLMs** (P1 = P2 = { LLaMa-3.1-8B, Qwen2.5-7B, Mistral-7B-v0.3}), and compare the questions generated via approximated (defined as $X_{\text{appro}}$) and exact (defined as $X_{\text{exac}}$ ) Eq.(1), respectively. We compared LLM value orientations obtained with $X_{\text{appro}}$ and $X_{\text{exac}}$, respectively, and observed:
> + Pearson correlation=0.86, suggesting that **the approximated Eq.(1) produces similar results**.
> + Cronbach's alpha=0.90, indicating that **the approximated Eq.(1) measures the same construct as the exact one**.
>
> Please refer to Appendix I.2 for more details.
>
> This empirical comparison provides strong evidence that **our approximations preserve the effectiveness of the method** and do not sacrifice its validity.
>
> ---
> ### **W4: Any reasons on your choice of hyperparameters? No experiments or searches provided.**
>
> As described in Sec.4.1 and Algorithm 1, AdAEM has the following hyperparameters.
>
> + Initial generic questions $X_1$ with size $N_1$: We directly apply all existing Schwartz value-related datasets we found.
> + Budget $B$: Control the optimization round and is determined by our available computation resource.
> + $N_2$: the number of new questions generated per exploration step. This balances quality and efficiency. We simply set it to a small, practical value ($N_2=3$).
> + $P_1$, $P_2$: LLMs to estimate the reward score during the optimization. Since AdAEM aims to explore value-eliciting questions by probing LLMs' value boundaries, the key criterion for selecting P1 and P2 is the potential diversity of their underlying values.
>
> **Most hyper-parameters are set by default following the above criteria, without an exhaustive search**.
>
> To further address the concern of hyperparameters' impact on AdAEM's performance, we conduct empirical robustness analysis.
>
> (1) **Initial question set $X_1$**: We randomly divided the questions into 5 distinct partitions, ran AdAEM separately, evaluated LLMs' values using the five resulting question sets, and compared the correlations between the values they produced. *Cronbach's $\alpha$=0.90 and Coefficient of Variation (CV) = 0.28*, **indicating consistent and reliable evaluation results without relying on specific test questions**. Details in Appendix K.3.
>
> (2) **Budget $B$**: Fig. 7 shows that the informativeness score monotonically increases with a larger $B$. We set $B$=1500, our maximum computational resources, to examine its convergence. Note that a high score can be achieved within only a moderate number of iterations.
>
> (3) **LLM participants $P_1, P_2$**: To further analyze AdAEM's robustness to $P_1,P_2$, we implement AdAEM with an alternative set of LLMs (P1={ LLaMa-3.1-8B, Qwen2.5-7B, Mistral-7B-v0.3, GPT-4o-Mini}=P2), and compare the generated questions (denoted as **AdAEM-2**) with the original AdAEM Bench.
>
> + *Question quality comparison*: both exhibit greater semantic diversity and topic richness than the manually crafted SVS, as shown below:
>
>     | **Dataset**   | **#q** | **Avg.L $\uparrow$** | **SB $\downarrow$** | **Sim $\downarrow$** |
>     |---------------|--------|-----------|--------|---------|
>     | **SVS**       | 57     | 13.00     | 52.68  | 0.61    |
>     | **AdAEM**     | 12,310 | 15.11     | 13.42  | 0.44    |
>     | **AdAEM-2** | 8,452  | 15.35     | 14.36  | 0.45    |
>
> + *Question informativeness comparison*: The average informativeness scores: SVS:6.07, AdAEM: 6.99, AdAEM-2: 6.51 (Fig.20). Though more advanced LLMs yield slightly higher rewards, *both strongly outperform the generic initial questions and previous static benchmarks*.
> + *Correlation of evaluation results*: Using both sets to evaluate the values of LLMs, respectively, their evaluation shows high agreements: Intra-Class Correlation (ICC)=0.81, Pearson=0.80, Cronbach's $\alpha$=0.84, confirming consistent value assessments across model combinations.
>
>     More details are given in Appendix K.1.
>
> In summary, **AdAEM achieves stable and meaningful results with default hyperparameter choices and is robust to hyperparameter settings**.

---

> ### Author Response · Authors · 2025-11-22
> **Thank you for your insightful comments and constructive feedback. Response to Reviewer J1nE (3/3)**
>
> #### **Q1. Fig.2 can be polished**
> Thank you for the valuable suggestion. We have provided an improved version of Fig.2 in the revised pdf, in which we (i) explicitly highlight the iterative explore-and-optimize process, and (ii) add concrete notations and brief descriptions to better align the figure with the textual description.
>
> #### **Q2. It would be better to unify the notation in the main text, notation table and pseudocode.**
> Thank you for the helpful suggestion. We carefully reviewed all notations in the main text, notation table, and pseudocode to ensure consistency and remove any unexplained symbols.
>
> We also provided a complete, unified notation list Appendix (Table 5, page 26). We believe these revisions could substantially improve clarity, readability, and presentation quality.
>
> #### **Q3. The “Figure 15” in AdAEM Question Generation on page 23 probably should be “Figure 12”.**
> Sorry for this reference error. "Fig.15" in AdAEM Question Generation on page 23 should indeed be "Fig.12".
>
> Fig.12 shows the geographical coverage of the full AdAEM Bench optimized with multiple LLMs to highlight the diversity, while Fig.15 reports the geographic distribution of questions generated by different LLMs, which justifies the motivation and effectiveness of our framework.
>
> We have fixed it in the revised pdf.
>
> ---
> We hope our responses and additional results could address your concerns. We are more than willing to respond to any further questions regarding our methods and experiments.
>
> We would sincerely appreciate it if you could review our responses and kindly reconsider the assessment of our work.
>
> ---
> ### Reference
> [1] Bai et al., Training a Helpful and Harmless Assistant with Reinforcement Learning from Human Feedback. 2022.
>
> [2] Tie et al., A Survey on Post-training of Large Language Models. 2025.
>
> [3] Scherrer et al., Evaluating the Moral Beliefs Encoded in LLMs. NeurIPS 2023.
>
> [4] Ren et al., ValueBench: Towards Comprehensively Evaluating Value Orientations and Understanding of Large Language Models. ACL 2024.
>
> [5] Sorensen et al., Value Kaleidoscope: Engaging AI with Pluralistic Human Values, Rights, and Duties. AAAI 2025.
>
> [6] Agakov, F. V. (2005). Variational Information Maximization in Stochastic Environments (Doctoral dissertation, University of Edinburgh).
>
> [7] Wan et al., f-Divergence Variational Inference. NeurIPS 2020.
>
> [8] Colombo et al., A Novel Estimator of Mutual Information for Learning to Disentangle Textual Representations. ACL 2021.

---

> > ### Comment · Reviewer_J1nE · 2025-11-27
> >
> > Thank you for your explanation and clarification, which addresses most of my concerns. I have no other further questions.

---

### Official Review · Reviewer_SGLU · 2025-11-04

**Soundness:** 3
**Presentation:** 3
**Contribution:** 3
**Rating:** 8
**Confidence:** 3

**Summary:**

The paper proposes a value difference elicitation framework AdAEM. The framework iteratively generates new questions and calculates scores for conformity, coherence, value difference and semantic difference, extending prior value benchmarks with questions that better highlight the value boundaries between LLMs. The authors show that the generated questions are more diverse compared to prior value benchmarks and calculate the value differences of 16 LLMs, using their framework.

**Strengths:**

- The framework is novel, extensible, and generalizable
- The framework and the generated dataset of 12k questions should be useful for researchers exploring value differences in LLMs
- The writing is easy to follow
- There is substantial analysis aiming to validate the effectiveness of the benchmark
- Includes a discussion on how the authors think about values for LLMs

**Weaknesses:**

- A lot of the analysis is conditioned on the Schwartz Value Survey. Though in principle the approach could work on other value frameworks, a proof of concept on a different set of questions would’ve strengthened the generalization capability of the framework.
- The validity analysis claims construct validity but assumes o3-mini’s capability of generating text with a particular value present, which is not a given.

**Questions:**

NA

---

> ### Author Response · Authors · 2025-11-22
> **Thank you for your thoughtful reviews and suggestions. Response to Reviewer SGLU (1/3)**
>
> Thank you very much for your supportive review and valuable suggestions. Our responses to each weakness (W) are as follows. A revision of our paper has been uploaded, with the revised parts marked in blue.
>
> ---
> ### **W1: A proof of concept on a different question set and value framework**
>
> Our framework is theoretically applicable to any value system, and we instantiated it with the Schwartz value system as it's the most widely used one in the context of LLM value evaluation/alignment.
>
> To further validate AdAEM's generalization capability, following your suggestion, we also endow it with another popular value framework, **Moral Foundation Theory (MFT)** and generate a new question set, **AdAEM Bench-MFT**.
>
> We add the full implementation details, experimental settings, and results analysis in **Appendix J**, and summarize key findings below:
>
> + **Question Quality**: **The questions generated by AdAEM demonstrate consistently much better semantic diversity** than manually crafted MFT questionnaires or ValueBench, as shown in the Table below:
>
>     | **Dataset** | **#q** | **Avg.L $\uparrow$** | **SB $\downarrow$** | **Sim $\downarrow$** |
>     |-------------|--------|-----------|--------|---------|
>     | **MFQ**     | 30     | 11.57     | 24.38  | **0.5031**  |
>     | **VB**      | 66     | 11.97     | 26.06  | 0.5487  |
>     | **AdAEM Bench-MFT**   | **589**    | **15.6**      | **20.86**  | 0.5263  |
>
> + **Value Difference Elicitation**: Since the primary goal is to expose LLMs' underlying value difference, we further check AdAEM's such capability on MFT. We evaluate *GPT-4.1, Mistral-7B-v0.3, Llama-3.3-70B-Instruct, and DeepSeek-V3* with AdAEM Bench-MFT, Moral Foundations Questionnaire (MFQ) and ValueBench.
>
>     | **Metrics** | **MFQ** | **ValueBench** | **AdAEM Bench-MFT**|
>     |-------------|--------|-----------|--------|
>     | **Corr among LLMs** ↓    | 0.580     | 0.623     | **-0.095**  |
>     | **std of values** ↑   | 0.096     | 0.132     | **0.212**  |
>
>     As shown above, we find:
>     + We calculate the average Pearson correlation of evaluated values among different LLMs. Both MFQ and ValueBench cause very high correlations, which indicate *different models' value tendencies are implausibly similar* measured by them.
>     + We show the average standard deviation of LLM's tendency scores across the five foundations and find that MFQ and ValueBench show little differentiation, indicating that *neither successfully reveals LLM value differences*.
>
>     In comparison, **AdAEM leads to low (negative) value correlation among different LLMs and high distinguishability across the five foundations** (high std), better exposing more value differences and providing informative results (refer to Table 16 and Fig.19 in Appendix J.2 for detailed evaluation results and radar chart visualization).
>
> + **Evaluation Validity**: To verify that AdAEMBench-MFT can reflect the real tendency of LLMs over the five foundations, we also conduct the controlled value priming experiments on MFT using GPT-5, as we did in Fig.4.
>     | **Dimension** | **Baseline** | **Controlled** | **Improvement**↑ |
>     |---------------|--------------|----------------|-----------------|
>     | **Care**      | 74.88        | 98.31          | 31.29%          |
>     | **Fairness**  | 80.43        | 98.07          | 21.93%          |
>     | **Loyalty**   | 54.35        | 98.79          | 81.77%          |
>     | **Authority** | 57.25        | 98.07          | 71.30%          |
>     | **Sancity**   | 30.19        | 97.83          | 224.05%         |
>
>     As shown in the table above, when we explicitly prompt GPT-5 to reflect a specific foundation in its responses, **AdAEM successfully captures the significant increase towards the target foundation**.
>
> Taken together, the results above show that AdAEM generalizes effectively to Moral Foundation Theory, preserving its key advantages to generate diverse, value-evoking questions to uncover LLMs' value differences. **This strengthens that AdAEM is a general framework rather than one tied to a particular value theory.**
>
> ----
> ### **W2: o3-mini's capability of generating text with a particular value is not verified.**
>
> We respond to this concern from three aspects:
>
> 1. **Justification for using in-context learning to steer LLMs' values**. We control o3-mini to reflect the target value by providing carefully designed system message instructions. Such methods, known as **In-Context Alignment (ICA)**, **have been empirically validated and widely used** to steer diverse traits of LLMs, such as personas [1,2,3], personality [4,5,6] as well as values [7,8,9].

---

> ### Author Response · Authors · 2025-11-22
> **Thank you for your thoughtful reviews and suggestions. Response to Reviewer SGLU (2/3)**
>
> 2. **Validity of ICA-based value steering**. To verify that o3-mini indeed changes behaviors under such priming, we also validate the effect using ValueBench.
>
>     The results in the following table show the average shift in the controlled, relevant, and opposite values when we enhance each value dimension in the Schwartz value system.
>
>     | **Controlled Dimension**      | **Change on Target Value** | **Change on Same Group** | **Change on Opposite Group** |
>     |--------------------|----------------------------|--------------------------|------------------------|
>     | **Achievement**    | 66.67%                     | 17.98%                   | 0.73%                  |
>     | **Benevolence**    | 14.29%                     | 1.82%                    | -19.97%                |
>     | **Conformity**     | 45.45%                     | 25.38%                   | -32.46%                |
>     | **Hedonism**       | 22.78%                     | 19.42%                   | -19.61%                |
>     | **Power**          | 108.88%                    | 105.56%                  | -5.36%                 |
>     | **Security**       | 4.55%                      | 19.89%                   | -3.95%                 |
>     | **Self-direction** | 2.63%                      | 6.08%                    | -24.87%                |
>     | **Stimulation**    | 44.45%                     | 34.88%                   | -0.63%                 |
>     | **Tradition**      | 29.17%                     | 18.18%                   | -10.68%                |
>     | **Universalism**   | 1.82%                      | 11.43%                   | -4.33%                 |
>
>     As shown above, even though ValueBench is less discriminative than our AdAEM, we still observe scores on target and related (in the same group) values increase substantially (+34.1%) and moderately (+26.1%), respectively, while scores on conflicting values decrease (-12.1%), **indicating that our ICA method successfully controls the target value**.
>
> 3. **Using an LLM with stronger ICL capability**. To further resolve the concern of o3-mini's capability for ICA, we repeat the same experiment using the more advanced **GPT-5-thinking**.
>
>     | **Dimension**      |  **Change on Target Value** | **Change on Same Group** | **Change on Opposite** |
>     |--------------------|----------------------------|--------------------------|------------------------|
>     | **Achievement**    |  76.50%                     | 21.74%                   | -8.33%                 |
>     | **Benevolence**    |  89.88%                     | 9.70%                    | -10.30%                |
>     | **Conformity**     | 91.51%                     | 50.48%                   | -89.34%                |
>     | **Hedonism**       |  3189.47%                   | 14.03%                   | 9.52%                  |
>     | **Power**          |  226.78%                    | 42.30%                   | -34.02%                |
>     | **Security**       |  9.21%                      | 12.83%                   | -87.31%                |
>     | **Self-direction** | 70.44%                     |    9.01%                | -28.81%                  |
>     | **Stimulation**    |  17.95%                     | 34.28%                   | 9.68%                  |
>     | **Tradition**      |  159.07%                    | 36.37%                   | -90.35%                |
>     | **Universalism**   |  35.62%                     | 25.96%                   | -27.74%                |
>
>     As shown above, the results also reflect the expected value change: target value (+396.6%), values in the same group (+25.7%), and conflicting values (-35.7%), **further supporting the construct validity**.
>     Besides, we also compare the average value changes on the four values in Fig.4 when using o3-mini and GPT-5 as the backbone, respectively.
>     | Backbone LLM     |  **Change on Target Value** | **Change on Same Group** | **Change on Opposite** |
>     |--------------------|----------------------------|--------------------------|------------------------|
>     | **o3-mini**    |  23.1%                     | 14.0%                   | -40.5%                 |
>     | **GPT-5**    |  119.7%                     | 27.9%                    | -40.60%                |
>
>     We see that AdAEM sensitively captures value shifts regardless of which LLM is used for value priming. Stronger backbone models (with potentially better value-steering ability) induce larger value changes, which are likewise detected by our method.
>
> Taken together, these results support our conclusion of construct validity.
>
> More detailed results are added in Appendix C.12.

---

> ### Author Response · Authors · 2025-11-22
> **Thank you for your thoughtful reviews and suggestions. Response to Reviewer SGLU (3/3)**
>
> ### Reference:
> [1] Choi and Li, PICLe: Eliciting Diverse Behaviors from Large Language Models with Persona In-Context Learning. ICML 2024.
>
> [2] Moon et al., Virtual Personas for Language Models via an Anthology of Backstories. EMNLP 2024.
>
> [3] Luz de Araujo, Helpful assistant or fruitful facilitator? Investigating how personas affect language model behavior. 2025.
>
> [4] Jiang et al., Evaluating and inducing personality in pre-trained language models. NeurIPS 2023.
>
> [5] Jiang et al., PersonaLLM: Investigating the Ability of Large Language Models to Express Personality Traits. Findings of NAACL 2024.
>
> [6] Kang et al., Are the values of llms structurally aligned with humans? A causal perspective. Findings of ACL 2025.
>
> [7] Xu et al., Align on the Fly: Adapting Chatbot Behavior to Established Norms. 2023.
>
> [8] Huang et al., How Far Can In-Context Alignment Go?
> Exploring the State of In-Context Alignment. Findings of EMNLP 2024.
>
> [9] Lin et al., The Unlocking Spell on Base LLMs: Rethinking Alignment via In-Context Learning. ICLR 2024.

---

### Author Response · Authors · 2025-11-27
**The revision of our work has been uploaded. Looking forward to your futher feedback!**

Dear Reviewers:

We sincerely thank you for your thoughtful and constructive feedback. Following your suggestions, **we have revised our paper, with the revised parts marked in blue, and uploaded the revision PDF**, which includes *all supplementary discussions, implementation details, and additional results referenced in our earlier responses*. All corresponding sections/lines have been adjusted accordingly. We hope these additions and clarifications could facilitate a fair reassessment of our work.

We fully understand that you might be quite busy and **sincerely appreciate any further comments you may have during the discussion period**. We are more than happy to address any further concerns or questions.

Again, we are grateful for the time and effort you have devoted to our submission.

Best,

The authors

---

### Author Response · Authors · 2025-12-01
**General Response to New ACs Regarding the Main Concerns**

We sincerely thank all the reviewers and ACs for their efforts and valuable comments. In hopes of reducing the new AC's workload caused by the accident, we'd like to summarize the reviewers' flagged Weaknesses (W) into **five** key concerns and explain how we address them. *Our corresponding substantial clarifications and improvements have extended the updated paper by 9 full pages* (40 pages $\rightarrow$ 49 pages), with the revised parts marked in blue.

1. **AdAEM's Generalizability on Other Value Systems** (Reviewers SGLU-W1, 1aVD-W1, xTVE-W2). Besides the Schwartz value system, **following the three reviewers' requirement**, we also instantiated our method, AdAEM, with the Moral Foundation Theory, and analyzed AdAEM's efficacy through three aspects: *i) quality of generated questions, ii) value difference elicitation ability, and iii) evaluation validity* (full results added in Appendix J ), **demonstrating its generalization ability to other value systems**.

2. **Empirical Approximation of Mathematical Modeling** (Reviewers J1nE-W3, xTVE-W1). Both reviewers were concerned that our LLM-based approximation of the probabilistic modeling in Eq. (1) might affect AdAEM's efficacy. To address this, we (i) comprehensively analyze the source and necessity of the approximation; (ii) **implement Eq. (1) exactly** via open-source models and **show that the exact and approximate implementations yield very similar results** (Pearson correlation = 0.86) and **measure the same construct (Cronbach's $\alpha$ = 0.90)** (full results added in Appendix I ). These results, together with the good question quality, reliability, and validity under the approximation, strongly verify that *our approximation preserves AdAEM's effectiveness and validity*.


3. **Hyperparameter Choice Analysis** (Reviewers J1nE-W4, xTVE-W3). We provide detailed hyperparameter settings and selection criteria, together with additional experiments showing that: (i) **using different LLM participants (P1, P2)** for generation **yields similar question quality and informativeness, as well as consistent value assessments (ICC = 0.81, Cronbach's $\alpha$ = 0.84)**; (ii) AdAEM's performance is relatively insensitive to the number of questions, with 1k questions performing comparably to 12k; and (iii) the results are not sensitive to the choice of initial questions (full experiments in Appendix K). *These findings indicate that AdAEM is robust to hyperparameter settings and does not require an exhaustive search*.

4. **Reliability of o3-mini for Value Priming** (Reviewer SGLU-W2). To verify that our ICL method effectively steers o3-mini's values, we i) test it on an additional benchmark to show its values are successfully controlled (construct validity), and ii) replicate the value-priming experiments with the more capable GPT-5.1 Thinking model, observing successful value control and similar patterns. These findings support our conclusion (see Appendix C.12 for details).

5. **Convergence Guarantee of AdAEM** (Reviewer J1nE-W2). To show such a guarantee, we i) present convergence proof given by the original IM paper; ii) provide the empirical evidence that AdaEM's scores can monotonically increase and stabilize after more iterations (Appendix E.6).

**For other presentation issues**, we i) explain our motivation of value different maximization in response and Appendix G (J1nE-W1); ii) redraw the whole Fig.2 (J1nE-Q1); iii) unify the symbols and provide a detailed notation table (Table 5) (J1nE-Q2); iv) explain the details of human experts in response and Appendix C.10 (xTVE-Q2); v) add more discussions on misuse (1aVD-W2) in the Ethical Statement (page 19). We believe these easy-to-fix presentation issues would not impede the core contributions of our novel method and empirical results.

Please note that **Reviewer J1nE has confirmed most concerns were addressed and raised the score from 4 to 8 before the incident was widely known. Reviewer xTVE likewise confirmed on the 26th that the concerns were resolved**.

In summary, we believe these revisions, expansions, and additional results further improve our work and address all the concerns.  We would sincerely appreciate it if you could read the reviews and our responses, and kindly consider the assessment of our work.

Best regards,

The Authors

---

### Meta-Review · Area_Chair_7MA8 · 2026-01-09

**Summary:**

The main concerns affecting the decision:
- generalizability beyond the Schwartz Value Survey
- validity/reliability assumptions for black-box probability estimates
- methodological design/analysis (e.g., EM-like procedure guarantees, hyperparameter justification, and potential bias induced by divergence-maximization)

Overall, the rebuttal has resolved most concerns, leading to an oral acceptance.

**Reviewer Concerns:**

Addressed by rebuttal:
- generalizability beyond the Schwartz Value Survey: additional experiments/analysis on other value systems were added  (SGLU, 1aVD, xTVE)
- methodological design/analysis: additional details/analysis (e.g., EM-like procedure guarantees, hyperparameter justification, and potential bias induced by divergence-maximization) were added (J1nE)
- reproducibility details & evaluation fairness: additional discussions were done (xTVE)
- reliance on o3-mini: beyond o3-mini, GPT-5.1 Thinking model was tested (SGLU)

Still outstanding:
- none.

**Reviewer Scores:**

- SGLU (original score 8): likely stays at positive, as most concerns were resolved.
- J1nE (original score 4): likely no change, or small upward improvements
- 1aVD (original score 8): likely no change, as most concerns were resolved
- xTVE (original score 8): likely no change, as most concerns were cleared

---

### Decision · Program_Chairs · 2026-01-26

Accept (Oral)